# CHEMTHINKER: THINKING LIKE A CHEMIST WITH MULTI-AGENT LLMS FOR DEEP MOLECULAR INSIGHTS

## ABSTRACT

Molecular property prediction is vital in drug discovery and cheminformatics, yet many current models lack interpretability, making it difficult for experts to understand the rationale behind predictions. To address this, we introduce Chem-Thinker, a novel large language models (LLMs) multi-agent framework designed to effectively control the internal representations of concepts and functions within LLMs. ChemThinker emulates the way chemists approach molecular analysis by integrating insights from three perspectives: general molecular properties, data-driven analysis, and task-specific factors. Each perspective uses an agentic approach to stimulate the LLM's internal representations, enabling more targeted and interpretable outputs based on the problem at hand, akin to how stimuli trigger the brain's cognitive processes. By feeding representations from these three perspectives into a simple multi-layer perceptron (MLP), ChemThinker achieves superior performance, significantly outperforming existing baselines across multiple benchmarks. Furthermore, our framework provides interpretable insights into the molecular mechanisms driving the predictions, making it a practical tool for drug discovery and other cheminformatics applications.

## 1 INTRODUCTION

*"If I have seen further it is by standing on the shoulders of giants."*

– Isaac Newton

Molecular property prediction plays a crucial role in cheminformatics (Yang et al., 2019) and drug discovery (Drews, 2000; Zhang et al., 2021), enabling experts to estimate essential characteristics such as blood-brain barrier permeability, solubility, and toxicity (Wu et al., 2023). While numerous works (Hu et al., 2019; You et al., 2020; Wang et al., 2022; Stärk et al., 2022; Zhou et al., 2023) exist for this task, they often fall short in providing interpretable insights into the underlying molecular mechanisms. In real-world applications, simply delivering a prediction is rarely sufficient. Domain experts require a deeper understanding of the rationale behind the predictions— the "why"—to make informed decisions. Unfortunately, this critical aspect is often overlooked.

Moreover, current approaches (Xia et al., 2022; Liu et al., 2022; Luo et al., 2024; Rollins et al., 2024) primarily focus on identifying statistical patterns via extensive model training to predict molecular properties. However, this often overlooks the incorporation of established scientific knowledge, limiting the model's ability to be guided toward more informed and contextually relevant solutions. Rather than "learning from scratch," leveraging well-established scientific knowledge and frameworks offers a more reasonable and effective approach to tackling domain-specific problems like molecular property prediction. In other words, by "standing on the shoulders of giants," models can be directed towards more accurate and interpretable predictions, enhancing their practical utility.

Recently, large language model (LLM) agents have gained significant attention (Taylor et al., 2022; Dubey et al., 2024; OpenAI et al., 2023). With extensive training on vast amounts of data, these models have developed emergent abilities (Wei et al., 2022), enabling them to perform complex tasks such as role-playing and reasoning, at levels akin to human experts (Chen et al., 2024), such as chemists (Wang et al., 2024). In some cases, LLM agents can match or even surpass human expert

performance across a wide range of tasks, including medical question answering (Singhal et al., 2023) and chemical reasoning task (Mirza et al., 2024). Remarkably, Yao et al. (2022) demonstrate that integrating reasoning traces with task-specific actions can enhance model interpretability and performance, and the collaborative LLM agents can even further enhance the performance on intricate tasks (Hong et al., 2023; Li et al., 2024).

In this paper, we propose ChemThinker, a Multi-Agent LLM framework designed to extract deep molecular insights from three distinct perspectives: General Molecular Thinking, Intuition-Driven Thinking, and Task-Specific Thinking. Here analysis of each perspective is taken charge of a specific LLM agent. The thought components extracted from three perspectives are then fused as representations, which can be utilized in a simple multi-layer perceptron (MLP) classifier for downstream tasks. This three-perspective analysis framework mirrors how chemists approach real-world molecular problems. Chemists typically begin with a general molecular analysis, followed by applying intuition from past experience and data analysis of similar molecules to infer properties. Finally, they focus on the specific task, adapting their approach to meet the unique requirements of the problem or prediction. Similarly, ChemThinker follows this process to generate molecular insights, providing a detailed text report that enhances the framework's interpretability and deepens the understanding of its predictions.

We demonstrate ChemThinker's performance across eight diverse molecular property prediction tasks, achieving state-of-the-art results on most tasks while also offering deeper molecular insights. Additionally, ChemThinker exhibits adaptability and flexibility by leveraging the varying contributions of each thought component as molecular representation to address different tasks. Our contributions are summarized as follows: (1) We propose an LLM-based Multi-Agent framework, ChemThinker, which emulates the thought process of chemists. This framework defines a new paradigm for molecular representation learning to improve molecular property prediction. (2) ChemThinker demonstrates flexibility by dynamically adjusting the contributions of each thought component, depending on the task and backbone model, leading to improved prediction accuracy and generalization across multiple datasets. (3) Through comprehensive experiments on both open- and closed-source LLMs, Chem-Thinker achieves state-of-the-art results in various molecular property prediction tasks, showcasing the power of leveraging pretrained LLMs for scientific applications. (4) In addition to providing accurate predictions, ChemThinker offers deeper molecular insights, aligning with the reasoning process used by chemists, thus improving the transparency and interpretability of the predictions.

## 2 RELTAED WORK

The accurate prediction of molecular properties is a cornerstone of cheminformatics (Yang et al., 2019) and bioinformatics (Zhang et al., 2021), relying heavily on the computational representation of molecular structures. Several representation methods have been developed, including molecular graph, Extended-Connectivity Fingerprints (ECFP) (Rogers & Hahn, 2010), and string line annotations such as Simplified Molecular Input Line Entry System (SMILES). With the advancement of artificial intelligence, machine learning models have been extensively utilized for property prediction through two main approaches: traditional models and deep learning models. In the conventional approach, computed or handcrafted molecular fingerprints are fed into traditional machine learning models, such as random forests (Breiman, 2001), effectively capturing the relationship between molecular substructures and their associated properties (Axen et al., 2017; Jeon & Kim, 2019). However, they normally rely on predefined fingerprints and may not fully capture the complex patterns and interactions within molecular structures.

On the other hand, standard deep learning models aim to extract expressive representations of molecules in a data-driven manner. Recent works (Hu et al., 2019; You et al., 2020; Wang et al., 2022; Xia et al., 2022) leverage Graph Neural Networks (GNNs) to directly learn from molecular graphs. By learning hierarchical representations that capture both local and global structural information, they can uncover complex patterns and relationships within molecules. Additionally, other studies (Zhou et al., 2023; Luo et al., 2024; Zheng et al., 2024; Rollins et al., 2024) also combine a sequence-based model with other backbone architectures to capture both contextual relationships from sequential representations such as SMILES and structural features from GNNs. Nevertheless, these data-driven approaches are often limited by the quality and size of available datasets, as they typically require training from scratch. While these standard deep learning models excel at capturing complex patterns,

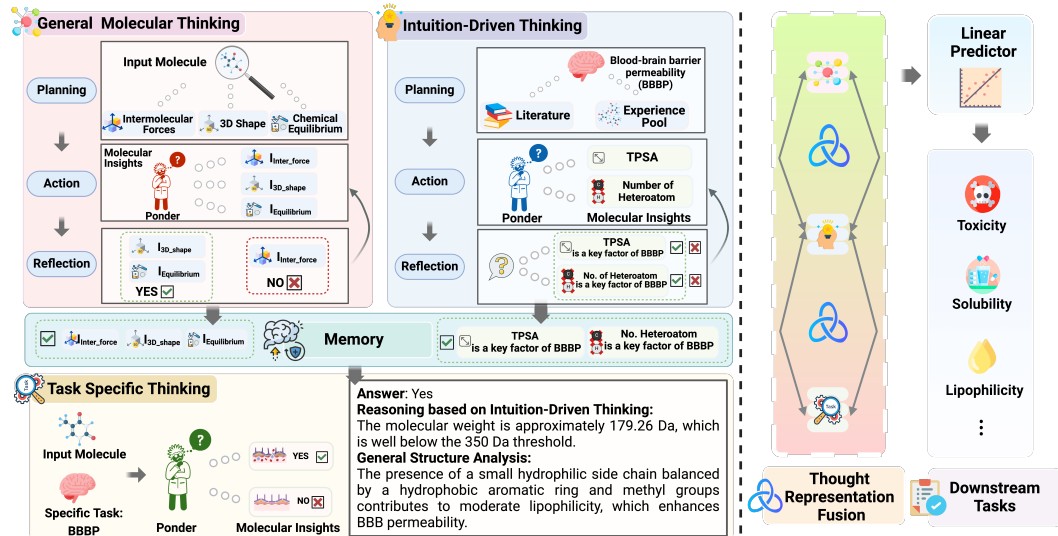

Figure 1: **The ChemThinker Framework.** The general molecular thinking workflow examines an input molecule's intermolecular forces, 3D structure, and chemical equilibrium. The intuition-driven thinking workflow retrieves relevant scientific rules from literature and the experience pool, i.e., related datasets, to help in addressing downstream tasks. The task-specific thinking workflow focuses on thinking of the final answer for the downstream task by utilizing the molecular insights provided by the first two agents stored in memory. After the agentic thinking processes, the thought representations generated by the three workflows are fused and then input into a linear predictor to produce the final predictions for various downstream tasks.

they often struggle to provide interpretable insights into the underlying molecular mechanisms, making it challenging to discern how underlying features contribute to the final predictions.

With the rapid advancement of Large Language Models (LLMs), remarkable success has been achieved across a wide array of tasks. Mirza et al. (2024) systematically analyze the chemical reasoning capabilities of LLMs, their findings indicate that even a LLM with 7B parameters can achieve an average human score, while more advanced models like GPT-4 (OpenAI et al., 2023) can surpass the highest human scores in chemical reasoning question answering. Furthermore, despite the growing adoption of LLMs in cheminformatics (Zhong et al., 2024; Jablonka et al., 2023; 2024; Liao et al., 2024), the potential of LLM-based Multi-Agent systems for molecular property prediction remains largely unexplored.

Therefore, to address these limitations, we propose a novel LLM-based Multi-Agent framework, ChemThinker, that leverages the collective intelligence, specialized profiles, and diverse skills of multiple agents. By standing on the shoulders of giants, our framework utilizes extensive pre-trained knowledge, human-like language generation, and reasoning abilities of LLMs. ChemThinker achieves more accurate molecular property predictions while generating detailed reports that experts could use for real-world analysis and decision-making.

## 3 CHEMTHINKER: THINKING LIKE A CHEMIST

In this section, we introduce ChemThinker, a novel multi-agent framework developed to extract deep molecular insights. As shown in Figure 1, ChemThinker leverages the collaborative capabilities of three agents from General Molecular Thinking, Intuition-Driven Thinking, and Task-Specific Thinking. For any given chemoinformatic task, ChemThinker harnesses its collective intelligence to generate a comprehensive report that provides deep molecular insights, along with fused representations to ensure precise downstream task predictions.

**Notation and Problem Definition.** We use The Simplified Molecular Input Line Entry System (SMILES) (Weininger, 1988) to represent molecular structures, where $S_i$ is a SMILES string and

$S$ is the SMILES dataset. The agent in each thinking component is represented as $A_{\text{Gen}}$, $A_{\text{Int}}$ and $A_{\text{Task}}$, each guided by the corresponding persona description $P$. Given a set of questions $\{q_1, q_2, \ldots, q_n\} \in Q$, the agent will provide a detailed analysis and explanation as the molecular insights $I$ for the task $T$, with each thinking component having its own representation **Rep**.

### 3.1 GENERAL MOLECULAR THINKING

In the general molecular thinking, $A_{\text{Gen}}$ is dedicated to examining and interpreting the overall architecture of molecular data, going beyond what traditional computational chemistry tools can achieve. While traditional tools excel at calculating explicit molecular properties such as molecular weight, ring counts, and basic geometries, they often fall short in extracting more nuanced insights that require a deeper contextual understanding. This is where the $A_{\text{Gen}}$ agent steps in, leveraging the advanced reasoning capabilities of LLMs to analyze complex molecular structures.

Based on a large amount pre-training corpora, $A_{\text{Gen}}$ excels in identifying subtle structural features, exploring relationships between functional groups, and understanding the broader implications of molecular configurations that are not readily accessible through conventional rule-based methods. In this thinking component, the $A_{\text{Gen}}$ initially identifies the key aspects of the molecular structure that would be most relevant to a chemist's real-world analysis. It then generates responses to the question $q_i$ for each aspect based on the input $S_i$, then proceeds to the reflection module to evaluate and correct its responses. The correct answer will stroed in the memory module. The questions $q_i \in Q_{\text{Gen}}$ as follows:

**Question 1** ($q_1$): *How does the molecule's 3D shape change, and what are the effects of these changes?*

**Question 2** ($q_2$): *What are the key intermolecular forces that govern the behavior of the molecule?*

**Question 3** ($q_3$): *How does the molecule contribute to the overall chemical equilibrium?*

For each question $q_i$, the agent $A_{\text{Gen}}$ generates corresponding molecular insights by analyzing the input $S_i$. The overall output of $A_{\text{Gen}}$, denoted as $I_{\text{Gen}}$, is defined as:

$$I_{\text{Gen}} = \{A_{\text{Gen}}(S_i, q_1), A_{\text{Gen}}(S_i, q_2), A_{\text{Gen}}(S_i, q_3)\} \tag{1}$$

This represents the set of molecular insights obtained from the agent's analysis of the given input and set of questions. In addition, the general structure thinking component includes a representation $\textbf{Rep}_{\text{Gen}}$, which captures the thinking process by analyzing three general questions and is subsequently used for downstream task prediction.

### 3.2 INTUITION-DRIVEN THINKING

To mimic how chemists use intuition built from domain knowledge and past experience, we introduce the intuition-driven thinking agent, $A_{\text{Int}}$. This agent is responsible for generating, refining, and applying heuristic rules that combine insights from literature with empirical observations from experience pools, i.e., related datasets to identify useful knowledge for a given task $T$. This is inspired by Zheng et al. (2023), which has shown that LLM is capable of extracting meaningful prior knowledge directly from literature and molecular patterns in the training data. By integrating computational tools and interacting with a coding agent, the $A_{\text{Int}}$ derives vectorized representations that are well-suited for further analysis and predictive modeling. The workflow in this thinking component consists of the following steps:

**Rule Generation from Literature Review:** LLM has built-in knowledge and an understanding of various tasks from its pre-training. To make use of this, we assign the LLM a specific persona, turning it into an agent that uses this persona to guide its behavior and thinking process. Based on this persona, the agent formulates a preliminary set of literature rules $R_{\text{lit}}$ relevant to the given task $T$. These rules are derived from the agent's internal understanding and pre-trained knowledge base.

**Rule Observation from Empirical Data:** Several randomly selected subsets of SMILES strings in the training data $\{S_i\}_{i=1}^m \in S_{\text{train}}$ with their corresponding label are then provided to the $A_{\text{Int}}$. By

analyzing these subsets, the agent derives additional rules $R_{\text{observed}}$ based on observed patterns and relationships within the molecular structures for the specific task $T$: $R_{\text{observed}} = \bigcup_{k=1}^{K} A_{\text{Int}}(\{S_i\}_{i=1}^{m}, T)_k$ where $K$ is the number of subsets analyzed. This step allows $A_{\text{Int}}$ to refine its rule set by incorporating specific data-driven insights from multiple perspectives, enhancing its understanding of the task $T$ and capturing context-specific nuances. After completing the reflection process, the refined rules, considered as molecular insight $I_{\text{Int}}$, will be stored in the memory module.

The final step involves utilizing the rule set $R_{\text{Int}} = R_{\text{lit}} \cup R_{\text{observed}}$ to generate numerical feature vectors for $S_i$. The $A_{\text{Int}}$ employs external tool such as RdKit (Landrum, 2013), and interacting to a coding agent to map each $S_i$ to a feature value $f_i$ based on the derived rules. This rule-based representation, encodes the domain-specific knowledge extracted by $A_{\text{Int}}$, will be used for downstream task prediction and is denoted as $\textbf{Rep}_{\text{Int}}$.

### 3.3 TASK SPECIFIC THINKING

The Task-Specific Thinking component utilizes $A_{\text{Task}}$ to directly answer specialized questions related to a given task $T$. Given the input $S_i$, the specific task $T$, and the molecular insights $I_{\text{Gen}}$ and $I_{\text{Int}}$ stored in the Memory Module, $A_{\text{Task}}$ synthesizes this information to generate task-specific insights. These insights are tailored to the specific requirements of $T$, leveraging the general and intuitive knowledge captured by the previous components. The resulting task-specific representation, denoted as $\text{Rep}_{\text{Task}}$, will be used to enhance the prediction accuracy for the downstream task. Furthermore, the molecular insights generated by this task-specific thinking component form the final report, which consolidates all molecular insights. This report can assist chemists for further analysis and decision-making. For reference, we also provide an example of a molecular insight report in the Appendix A.3.

### 3.4 THOUGHT REPRESENTATION FUSION

This representation fusion module integrates the diverse representations of $S_i$ generated from the three distinct modules: General Molecular Thinking, Intuition-Driven Thinking, and Task-Specific Thinking. The aim is to create a unified, context-aware representation of the molecular structure that captures the unique insights derived from each cognitive perspective, dynamically adapting to each input.

Given that current LLM architectures are based on next-token prediction, the observations produced in response to a question are inherently influenced by the question's context. For the General Molecular Thinking and Task-Specific Thinking module, the question serves as a guide to generate a representation for SMILES $S_i$. This representation acts as the thought component representation, encapsulating the LLM's thinking process for the molecule under each specified context.

Specifically, in the General Molecular Thinking, three different questions $(q_{\text{Gen1}}, q_{\text{Gen2}}, q_{\text{Gen}_3}) \in Q_{\text{Gen}}$ are posed to the LLM to assess various aspects of the molecular structure. Each question generates a separate embedding, which are then concatenate into a single, aggregated representation for general component representation:

$$\textbf{Rep}_{\text{Gen}}(S_i) = \textbf{Rep}_{\text{Gen}}(S_i, q_{\text{Gen}_1}) \oplus \textbf{Rep}_{\text{Gen}}(S_i, q_{\text{Gen}_2}) \oplus \textbf{Rep}_{\text{Gen}}(S_i, q_{\text{Gen}_3}) \tag{2}$$

In the Intuition-Driven Thinking, the representation is constructed using rule-based features derived from the empirical analysis of the molecule. Each of these rule values provides a unique view of the SMILES representation from a specific aspect. Specifically, for each SMILES string $S_i \in S$, each rule $r_j \in R$ is used to compute a feature value $f_{ij}(S_i)$, forming the feature representation $\textbf{Rep}_{\text{Int}}$ for $S_i$:

$$\textbf{Rep}_{\text{Int}}(S_i) = \{f_{i1}(S_i), f_{i2}(S_i), \ldots, f_{in}(S_i)\} \tag{3}$$

where $f_{ij}(S_i)$ represents the feature value for $S_i$ based on rule $r_j$, and $n$ is the total number of rules in $R$.

To generate the final unified representation, a dynamic weight vector $\mathbf{w}_i = (w_{\text{Gen},i}, w_{\text{Int},i}, w_{\text{Task},i})$ is calculated for each SMILES input $S_i$, reflecting the relative importance of each thinking component

for that specific molecule. The final fused representation is obtained by the weighted sum of the individual embeddings:

$$\mathbf{Rep}_{\text{fusion}}(S_i) = w_{\text{Gen},i} \cdot \mathbf{Rep}_{\text{Gen}}(S_i) + w_{\text{Int},i} \cdot \mathbf{Rep}_{\text{Int}}(S_i) + w_{\text{Task},i} \cdot \mathbf{Rep}_{\text{Task}}(S_i)$$

where $\mathbf{Rep}_{\text{Task}}(S_i) = \mathbf{Rep}_{\text{Task}}(S_i, q_{\text{Task}})$, and the dynamic weights $w_{\text{Gen},i}$, $w_{\text{Int},i}$, and $w_{\text{Task},i}$ are specific to each SMILES $S_i$ and are learned during the training process. These weights adaptively adjust based on the context and characteristics of the input molecule, ensuring that the final representation $\mathbf{Rep}_{\text{fusion}}(S_i)$ captures the most relevant information from each thinking mode.

The fused embedding $\mathbf{Rep}_{\text{fusion}}(S_i)$ is then passed through a Multi-Layer Perceptron (MLP) to generate the final prediction:

$$\hat{y}_i = \text{MLP}(\mathbf{Rep}_{\text{fusion}}(S_i))$$

where $\hat{y}_i$ represents the predicted outcome for the SMILES input $S_i$. The MLP is trained to map the comprehensive information encoded in $\mathbf{Rep}_{\text{fusion}}(S_i)$ to the desired prediction task, leveraging the full scope of insights derived from the fused representation. Therefore, ChemThinker allows for a flexible and context-aware integration of insights from different cognitive modules, dynamically adapting to the unique characteristics of each SMILES input.

## 4 EXPERIMENTS

### 4.1 EXPERIMENTAL SETUP

**Datasets.** Our framework is evaluated on 8 datasets across three chemoinformatics domains, covering 34 subtasks sourced from MoleculeNet (Wu et al., 2018). In the physiology domain, we address 29 tasks, including the blood-brain barrier permeability predictor (BBBP) (Martins et al., 2012), the clinical toxicity evaluator (ClinTox) (Gayvert et al., 2016), and 27 tasks from SIDER (Kuhn et al., 2016) focusing on predicting adverse drug reactions. In the biophysics domain, we evaluate 2 classification tasks from BACE (Subramanian et al., 2016) and HIV (Wu et al., 2018), while the physical chemistry domain covers 3 regression tasks from ESOL (Delaney, 2004), FreeSolv (Mobley & Guthrie, 2014), and Lipophilicity (Wu et al., 2018). To ensure fair comparisons across baselines, we employed scaffold splitting, following the MoleculeNet-recommended data split (Wu et al., 2018), which assigns molecules with different structural scaffolds to separate training, validation, and test sets (Bemis & Murcko, 1996).

**Baselines.** For the traditional model, we use Random Forest (Breiman, 2001) with ECFP4 (Rogers & Hahn, 2010) as the input feature set. For standard deep learning models, we selected the most representative Graph Neural Networks (GNNs) pretraining baselines, including Attribute Masking (AttrMask) (Hu et al., 2019), GraphCL (You et al., 2020), MolCLR (Wang et al., 2022), and MoleBERT (Xia et al., 2022). Additionally, we also compared with models like GraphMVP (Liu et al., 2022), 3D-InfoMax (Stärk et al., 2022), and Uni-Mol (Zhou et al., 2023), which leverage 3D molecular conformers during training to enhance their ability to reason in 3D space, while using 2D molecular graphs as input during inference. In addition, We also compare our framework with LLM4SD (Zheng et al., 2023) framework, which uses LLM generated rules as vectors and then utilizes a Random Forest for prediction. To ensure a fair comparison, we reran all baseline models using the same data-splitting method and the same random seed for multiple iterations. This consistency across runs ensures that any differences in performance are attributable to model variations rather than random factors.

**Backbone Model.** Our framework is built on an LLM-based Multi-Agent architecture, utilizing state-of-the-art LLMs as its backbone. Specifically, we utilize Galactica models (Taylor et al., 2022), with 6.7B and 30B parameters, which are trained on an extensive corpus of scientific knowledge. Additionally, we incorporate LLama-3.1 models (Dubey et al., 2024), including the 8B parameter version and its instruction-tuned variant (8B-instruct), both pretrained on a diverse dataset up to the end of 2023. For the closed-source model, we employ OpenAI's text embedding models (OpenAI, 2024) in both small and large configurations, alongside GPT-4o (OpenAI et al., 2023) for agent reasoning.

## 4.2 EXPERIMENTAL RESULTS

| | Dataset Model | Backbone Type | BBBP(1) ↑ | BACE(1) ↑ | ClinTox(1) ↑ | HIV(1) ↑ | SIDER(27) ↑ |
|---|---|---|---|---|---|---|---|
| | **RF + ECFP4** (Rogers & Hahn, 2010) | RF | 67.6 ± 1.0 | **85.0 ± 1.2** | 69.4 ± 3.1 | 77.1 ± 0.7 | 62.5 ± 0.5 |
| Baselines | **AttrMask** (Hu et al., 2019) | GNN | 65.2 ± 1.4 | 77.8 ± 1.8 | 73.5 ± 4.3 | 75.3 ± 1.5 | 60.5 ± 0.9 |
| | **GraphCL** (You et al., 2020) | GNN | 67.8 ± 2.4 | 74.6 ± 2.1 | 77.5 ± 3.8 | 75.1 ± 0.7 | 59.8 ± 1.3 |
| | **GraphMVP**(Liu et al., 2022) | GNN | 70.8 ± 0.5 | 79.3 ± 1.5 | 79.1 ± 2.8 | 76.0 ± 0.1 | 60.2 ± 1.1 |
| | **3D-infomax** (Stärk et al., 2022) | GNN | 69.1 ± 1.2 | 78.6 ± 1.9 | 62.7 ± 3.3 | 76.1 ± 1.3 | 56.8 ± 2.1 |
| | **MolCLR** (Wang et al., 2022) | GNN | 73.1 ± 1.6 | 81.5 ± 1.6 | 91.6 ± 2.7 | 77.3 ± 1.3 | 59.6 ± 0.7 |
| | **MoleBert** (Xia et al., 2022) | GNN | 71.9 ± 1.6 | 80.8 ± 1.4 | 78.9 ± 3.0 | 78.2 ± 0.8 | 62.8 ± 1.1 |
| | **Uni-Mol** (Zhou et al., 2023) | Transformer | 71.5 ± 1.4 | **84.4 ± 2.1** | 87.8 ± 2.6 | 78.3 ± 1.3 | 62.3 ± 5.6 |
| | **LLM4SD** (Zheng et al., 2023) | LLM | 74.5 ± 0.2 | 83.8 ± 0.3 | 92.8 ± 0.5 | **79.0 ± 0.2** | **64.8 ± 1.4** |
| | **ChemThinker**($Llama_{\text{Instruct}}$) | LLM | **77.0 ± 1.0** | 77.4 ± 1.9 | **99.1 ± 0.4** | 75.92 ± 1.1 | 62.7 ± 0.4 |
| | **ChemThinker**($OpenAI_{\text{Large}}$) | LLM | **75.5 ± 1.3** | 78.2 ± 0.9 | **99.4 ± 0.1** | **79.5 ± 0.7** | **63.7 ± 0.3** |

Table 1: Results on molecular property classification tasks with scaffold split. Mean and standard deviation of the **ROC-AUC (%)** performance from 10 random seeds are reported, with higher scores indicating better performance. The top-2 performances on each dataset are shown in bold, with **bold** being the best result, and *bold* being the second best result.

### 4.2.1 OVERALL PERFORMANCE

We evaluate ChemThinker on five classification datasets with 31 subtasks for molecular property prediction, and three regression tasks, as shown in Tables 1 and 2. For both tasks, we report the mean and standard deviation from 10 random seeds: ROC-AUC (%) for classification, where higher scores indicate better performance, and RMSE for regression, where lower values signify better result. We present ChemThinker results with two top-performing backbone models for both classification and regression tasks, compared against other state-of-the-art (SOTA) baselines.

| | Dataset Model | ESOL(1) ↓ | FreeSolv(1) ↓ | Lipophilicity(1) ↓ |
|---|---|---|---|---|
| | **RF + ECFP4** (Rogers & Hahn, 2010) | 1.34 ± 0.01 | 4.36 ± 0.04 | 0.90 ± 0.00 |
| Baselines | **AttrMask** (Hu et al., 2019) | 1.11 ± 0.05 | 2.92 ± 0.03 | 0.73 ± 0.00 |
| | **GraphCL** (You et al., 2020) | 1.31 ± 0.07 | 3.60 ± 0.32 | 0.78 ± 0.02 |
| | **GraphMVP**(Liu et al., 2022) | 1.06 ± 0.02 | 2.95 ± 0.19 | 0.69 ± 0.01 |
| | **3D-InfoMax** (Stärk et al., 2022) | 0.89 ± 0.04 | 2.83 ± 0.10 | 0.70 ± 0.02 |
| | **MoleBert** (Xia et al., 2022) | 1.02 ± 0.03 | 3.08 ± 0.05 | 0.68 ± 0.02 |
| | **MolCLR** (Wang et al., 2022) | 1.31 ± 0.03 | 2.73 ± 0.08 | 0.74 ± 0.02 |
| | **Uni-Mol** (Zhou et al., 2023) | 1.55 ± 0.26 | 3.94 ± 0.50 | 1.19 ± 0.07 |
| | **LLM4SD** (Zheng et al., 2023) | 0.52 ± 0.04 | 2.62 ± 0.01 | *0.68 ± 0.00* |
| | **ChemThinker**($Llama_{\text{Instruct}}$) | **0.44 ± 0.01** | **2.01 ± 0.37** | 0.73 ± 0.03 |
| | **ChemThinker**($Galactica_{\text{6.7b}}$) | *0.53 ± 0.25* | *2.39 ± 1.39* | **0.66 ± 0.02** |

Table 2: Results on Molecular Property Regression tasks with Scaffold Split. Mean and standard deviation of the **the Root Mean Square Error (RMSE) metric** from 10 random seeds are reported, with lower scores indicating better performance. The top-2 performances on each dataset are shown in bold, with **bold** being the best result, and *bold* being the second best result.

As shown in Table 1, Our framework demonstrates superior performance, particularly on the BBBP, ClinTox, and HIV datasets, where it surpasses existing baselines with notable improvements over both LLM and GNN-based frameworks. Notably, our framework exhibits exceptional performance on the Clintox dataset, achieving a near-perfect accuracy of 99.4% and 99.1%, this result significantly outperforms all other models. Moreover, on the SIDER datset, our model attains an score of 63.7%, which is comparable to the best perfoming LLM4SD (Zheng et al., 2023) at 64.8%, and superior to all GNN-based baselines. This result further enhances the credibility of LLM-based approaches in molecular property prediction tasks. In the case of the BACE dataset, ChemThinker achieves 78.2, which, while competitive, remains below the highest baseline result of 85.0 achieved by the RF model. The potential reasons may be the BACE dataset assigns binary labels for molecular inhibitors of human $\beta$-secretase 1 (BACE-1), based on an arbitrary threshold of quantitative potency values ($IC_{50}$) set at 7 (Wu et al., 2018). However, potency values can vary significantly depending on the assay settings (Landrum & Riniker, 2024), lower potency values can still indicate strong inhibition of BACE-1 (Harding et al., 2024). We hypothesize that this arbitrary threshold and label ambiguity hinder LLM agents' ability to reason effectively.

The molecular property regression results presented in Table 2 demonstrate the superior performance of our proposed framework compared to all baselines across three datasets: ESOL, FreeSolv, and Lipophilicity. ChemThinker achieves RMSE values of 0.44 and 0.53 on the ESOL dataset, and 2.01 and 2.39 on FreeSolv, outperforming all LLM-based and GNN-based backbone models. On the Lipophilicity dataset, ChemThinker with the Galactica-6.7bvariant achieves SOTA results with an RMSE of 0.66, while our variant with Llama-3.1-Instruct embedding backbone demonstrates competitive performance against other baseline models.

## 4.3 EFFECTIVENESS OF THOUGHT REPRESENTATION

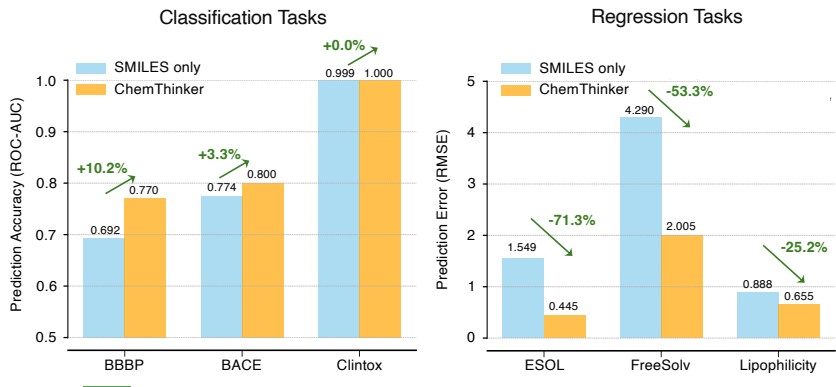

Figure 2: Comparison of the performance between ChemThinker and the SMILES-only representation across 10 random seeds on six datasets.

To assess the effectiveness of our proposed thought representation, we also evaluate SMILES only representation setting, as shown in Figure 2, the LLM used for encoding the SMILES-only representation is the best-performing backbone model from our proposed framework. The comparison with other LLM backbone models can be found in Appendix A.2, where we observe a similar trend across these six datasets. As shown in Figure 2, our proposed method consistently improved the scores across all six datasets. Notably, on the FreeSolv regression dataset, where a lower RMSE indicates better performance, we achieved a significant reduction in prediction error from 4.29 to 2.01. This improvement underscores the effectiveness of our framework, particularly in molecular property regression tasks. Similarly, for other regression and classification tasks, such as ESOL and BBBP, our method demonstrated a measurable improvement compared to the SMILES-only baseline.

In the Clintox dataset, the SMILES-only baseline achieved nearly perfect results, with our proposed method offering only marginal improvement. This may be due to the fact that the LLM already possesses sufficient understanding of the relevant domain and SMILES structure analysis. This further demonstrates that LLMs are more effective than other standard sequence encoders in learning string-based annotations for molecular structures.

## 4.4 THOUGHT COMPONENT CONTRIBUTION ANALYSIS

In this section, we analyze the contribution of each thinking component to the final decision, as illustrated in Figure 3. The evaluation is conducted on three classification datasets (BACE, BBBP, and Clintox) and three regression tasks (ESOL, FreeSolv, and Lipophilicity). Interestingly, the General molecular thinking component contributes more significantly to the classification tasks, whereas the Intuition-Driven Thinking component and Task-Specific component play a larger role in the regression tasks. This suggests that classification tasks may benefit from broader, more generalized features, while regression tasks require more domain-specific and intuitive insights to achieve optimal performance.

From the LLM architecture perspective, we find that LLMs with the same architecture tend to exhibit similar component contributions across different datasets to the final predictions. However, in ESOL and FreeSolv datasets, even though the LLMs heavily rely on one or two components, Galactica-6.7B and 30B demonstrate different behavior, despite having the same architectural design, these

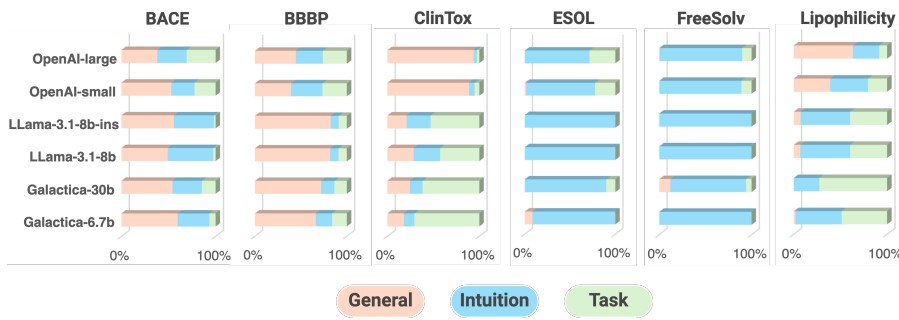

Figure 3: Thought Component Contribution Analysis. General, Intuition, and Task are short for General Molecular Thinking, Intuition-Driven Thinking, and Task-Specific Thinking, respectively. Each component contribution to the final score is calculated by averaging component weights for each SMILES across 10 random seeds.

models exhibit different component contributions for their final decisions. This pattern highlights the flexibility and adaptability of our proposed multi-agent framework.

Furthermore, the results for the ClinTox dataset, as reported in Table 1, demonstrate that we achieve near-perfect scores across all backbone settings. However, this component analysis reveals intriguing insights into how different model architectures rely on various components to make their predictions. The OpenAI models heavily depend on the General Molecular component, indicating that their decision-making process is primarily driven by general understanding and extensive pre-trained knowledge. In contrast, the Galactica models rely more on task-specific components, likely because our task-specific thinking process is closely aligned with their pre-training dataset and methodology. The LLaMA-3.1 models demonstrate a relatively balanced utilization of all three components, relying on every component across the general, intuition, and task-specific components to make accurate predictions.

### 4.4.1 EVALUATION OF DIFFERENT BACKBONE MODELS

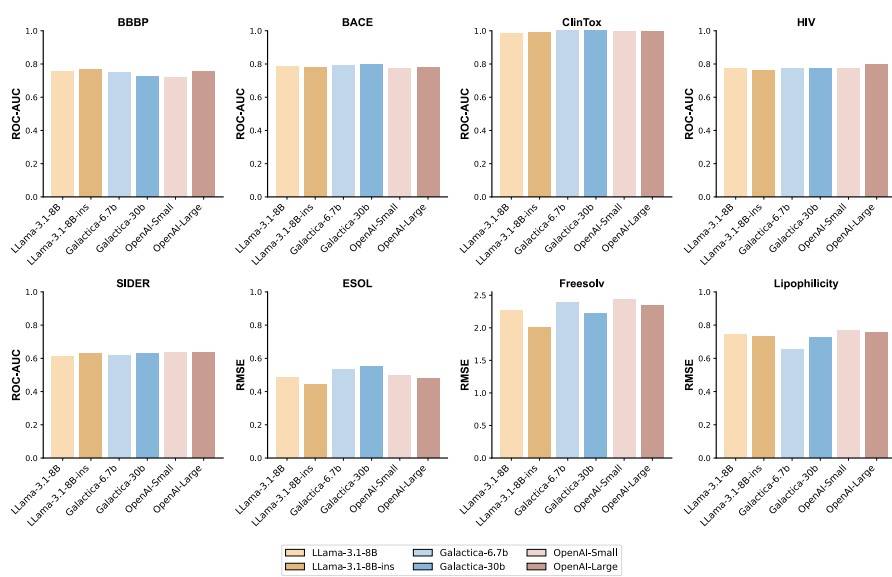

Figure 4: Evaluation of Different Backbone Models on 5 classification datasets and 3 regression datasets.

Figure 4 illustrates the evaluation of ChemThinker with different backbone LLMs. In the classification datasets, all models achieve high ROC-AUC scores with minimal differences, indicating

stable performance of our proposed framework across various tasks. For the regression datasets (ESOL, FreeSolv, Lipophilicity), Llama-3.1-8B-ins achieves the lowest RMSE, reinforcing its strong generalization ability. OpenAI-Large, on the other hand, tends to perform less favorably in these tasks, particularly for FreeSolv. Our proposed framework, especially with Llama-3.1-8B-instruct, excels with the lowest RMSE scores, demonstrating its strong ability to generalize and make accurate predictions even in more challenging tasks.

## 5    DISCUSSION

Our proposed LLM-based Multi-Agent framework, ChemThinkier, not only aims to predict molecular properties with greater accuracy but also seeks to provide deeper insights into the rationale behind these predictions. Inspired by the approach chemists take when addressing molecular problems, we designed three distinct modules: general molecular thinking, intuition-driven thinking, and task-specific thinking. Each module generates corresponding molecular insights, offering a more comprehensive understanding of the molecule. This multi-perspective approach allows ChemThinker to not only enhance prediction performance but also improve interpretability, making it highly valuable for both scientific discovery and practical applications. In the Appendix A.3, we provide a detailed example using a SMILES string, showcasing the ultimate molecular insights generated by each module.

We find that while reports generated by open-source models provide significant value, they tend to be less coherent and transparent in conveying their underlying reasoning. Even though open-source LLMs may not express their "thinking" process as effectively as closed-source models, they still possess a sufficient ability to generate representations that address prediction tasks well. Perhaps the gap in report generation quality is due to the lack of access to proprietary data and investment in instruction-based fine-tuning. Due to their pre-training on large-scale corpora, models like Galactica (Taylor et al., 2022), which is trained on extensive scientific knowledge, and Llama-3.1 (Dubey et al., 2024), which leverages more recent datasets, demonstrate a robust foundational understanding. Moreover, our experiments show that, in certain tasks, these open-source LLMs can achieve state-of-the-art (SOTA) results, further highlighting their capabilities in molecular analysis.

During the molecular insights generation and reflection process, LLMs tend to produce redundant responses, and controlling the generated length can be challenging. If we include more content encoding in the molecular representation, critical information such as the original SMILES may receive less attention during the generation process. To mitigate this, we limit the input to the question along with its corresponding SMILES string. Another reason for this approach is that LLMs rely on the next-token prediction, meaning the quality of their responses is heavily influenced by the input query and how effectively the model embeds it. We believe that, compared to the generation process itself, the more critical aspect is how LLM "think" or process the question. Thus, we designed three thinking modules to emulate the way chemists approach molecular analysis. Our experimental results demonstrate that ChemThinker is adaptable in leveraging varying contributions from each thought component to address different tasks, depending on the backbone model used. This flexibility allows ChemThinker to optimize its process to the specific requirements of each task, enhancing its overall effectiveness in molecular property prediction.

## 6    CONCLUSION

In this paper, we introduce, ChemThinker, our proposed LLM-based Multi-Agent framework that demonstrates significant flexibility and effectiveness in molecular property prediction by adapting its thought components to different tasks and backbone models. Standing on the shoulders of giants, we harness the strong reasoning capabilities, extensive pretrained knowledge, and encoding ability of LLMs to provide both accurate predictions and deeper insights into the reasoning behind those predictions. Our experimental results show that this approach not only achieves state-of-the-art performance in certain tasks but also enhances the interpretability of molecular analysis, aligning with the way chemists approach problem-solving in the field. Future work could explore additional tasks and architectures to further extend ChemThinker's applicability and performance.

## 7 ETHICS STATEMENT

Our work focuses solely on scientific challenges and does not involve human subjects, animals, or environmentally sensitive materials. We foresee no ethical risks or conflicts of interest. We are committed to upholding the highest standards of scientific integrity and ethical conduct to ensure the validity and reliability of our findings.

## 8 REPRODUCIBILITY STATEMENT

Our model is clearly formalized in the main text for clarity and comprehensive understanding. Detailed implementation, including dataset information, baselines, experimental settings, and model configurations, are provided in Section 3 and Section 4. The experimental settings and baselines have been rigorously checked for fair comparison. Our code will be made public upon acceptance.

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
