# A  APPENDIX

## A.1  FULL RESULTS ON 27 TASKS FOR THE SIDER DATASET

| Sider (↑) | llama-3.1 | | Galactica Series | | OpenAI | |
|---|---|---|---|---|---|---|
| | llama-3.1-8B | llama-3.1-8B-instruct | galactica-6.7b | galactica-30b | Small | Large |
| cardiac disorders | 64.70 ± 0.04 | 64.78 ± 0.04 | 62.50 ± 0.03 | 63.63 ± 0.03 | 62.07 ± 0.02 | 62.96 ± 0.03 |
| neoplasms benign, malignant and unspecified (incl cysts and polyps) | 55.97 ± 0.03 | 57.06 ± 0.03 | 59.96 ± 0.04 | 60.35 ± 0.03 | 61.82 ± 0.02 | 63.19 ± 0.02 |
| hepatobiliary disorders | 57.09 ± 0.03 | 62.02 ± 0.03 | 63.32 ± 0.03 | 63.41 ± 0.03 | 60.39 ± 0.02 | 58.27 ± 0.02 |
| metabolism and nutrition disorders | 54.61 ± 0.02 | 55.36 ± 0.03 | 55.25 ± 0.02 | 51.85 ± 0.02 | 57.97 ± 0.02 | 54.66 ± 0.02 |
| product issues | 67.43 ± 0.14 | 66.85 ± 0.12 | 60.79 ± 0.11 | 66.60 ± 0.11 | 63.00 ± 0.06 | 73.50 ± 0.06 |
| eye disorders | 58.26 ± 0.03 | 59.14 ± 0.02 | 58.00 ± 0.03 | 59.27 ± 0.03 | 66.33 ± 0.03 | 69.47 ± 0.03 |
| investigations | 63.80 ± 0.06 | 66.57 ± 0.06 | 62.53 ± 0.04 | 60.74 ± 0.04 | 62.53 ± 0.04 | 75.76 ± 0.04 |
| musculoskeletal and connective tissue disorders | 60.42 ± 0.03 | 64.86 ± 0.05 | 63.13 ± 0.03 | 68.59 ± 0.03 | 64.24 ± 0.03 | 68.94 ± 0.02 |
| gastrointestinal disorders | 65.68 ± 0.04 | 66.74 ± 0.04 | 67.95 ± 0.05 | 71.16 ± 0.03 | 71.81 ± 0.04 | 61.50 ± 0.04 |
| social circumstances | 59.24 ± 0.03 | 56.07 ± 0.03 | 53.14 ± 0.04 | 53.34 ± 0.04 | 55.29 ± 0.04 | 59.18 ± 0.03 |
| immune system disorders | 62.75 ± 0.02 | 62.84 ± 0.03 | 58.61 ± 0.02 | 61.73 ± 0.03 | 63.03 ± 0.03 | 61.43 ± 0.03 |
| reproductive system and breast disorders | 66.16 ± 0.03 | 67.11 ± 0.02 | 64.65 ± 0.04 | 67.45 ± 0.03 | 73.53 ± 0.03 | 70.72 ± 0.03 |
| general disorders and administration site conditions | 57.87 ± 0.07 | 59.72 ± 0.08 | 72.48 ± 0.06 | 65.98 ± 0.04 | 55.14 ± 0.04 | 71.21 ± 0.04 |
| endocrine disorders | 60.60 ± 0.03 | 59.23 ± 0.03 | 68.71 ± 0.03 | 68.96 ± 0.03 | 71.79 ± 0.02 | 56.86 ± 0.04 |
| surgical and medical procedures | 56.95 ± 0.06 | 56.15 ± 0.07 | 58.51 ± 0.05 | 63.14 ± 0.04 | 53.76 ± 0.03 | 53.73 ± 0.04 |
| vascular disorders | 58.03 ± 0.04 | 56.59 ± 0.03 | 59.23 ± 0.04 | 56.34 ± 0.03 | 57.92 ± 0.03 | 60.71 ± 0.03 |
| blood and lymphatic system disorders | 66.64 ± 0.04 | 76.29 ± 0.04 | 73.04 ± 0.03 | 72.33 ± 0.03 | 73.40 ± 0.02 | 72.13 ± 0.03 |
| skin and subcutaneous tissue disorders | 77.53 ± 0.08 | 76.68 ± 0.09 | 61.90 ± 0.09 | 58.69 ± 0.07 | 58.59 ± 0.04 | 59.83 ± 0.06 |
| congenital, familial and genetic disorders | 59.01 ± 0.04 | 61.69 ± 0.03 | 59.19 ± 0.03 | 62.88 ± 0.04 | 64.68 ± 0.03 | 65.23 ± 0.03 |
| infections and infestations | 63.22 ± 0.02 | 63.97 ± 0.03 | 62.97 ± 0.02 | 63.96 ± 0.02 | 63.20 ± 0.03 | 61.58 ± 0.03 |
| respiratory, thoracic and mediastinal disorders | 64.67 ± 0.03 | 64.95 ± 0.04 | 64.00 ± 0.03 | 62.86 ± 0.03 | 59.60 ± 0.03 | 63.94 ± 0.03 |
| psychiatric disorders | 58.99 ± 0.03 | 55.61 ± 0.03 | 58.84 ± 0.03 | 56.48 ± 0.03 | 63.94 ± 0.03 | 67.14 ± 0.03 |
| renal and urinary disorders | 59.53 ± 0.03 | 62.42 ± 0.02 | 64.17 ± 0.03 | 60.83 ± 0.03 | 76.31 ± 0.03 | 66.75 ± 0.03 |
| pregnancy, puerperium and perinatal conditions | 64.33 ± 0.06 | 67.32 ± 0.07 | 56.87 ± 0.06 | 65.77 ± 0.05 | 61.29 ± 0.03 | 54.64 ± 0.05 |
| ear and labyrinth disorders | 51.90 ± 0.02 | 54.79 ± 0.02 | 54.12 ± 0.02 | 53.64 ± 0.03 | 58.36 ± 0.02 | 53.42 ± 0.02 |
| nervous system disorders | 66.43 ± 0.06 | 70.90 ± 0.05 | 69.05 ± 0.05 | 70.94 ± 0.04 | 78.07 ± 0.05 | 70.63 ± 0.05 |
| injury, poisoning and procedural complications | 55.56 ± 0.03 | 57.37 ± 0.03 | 57.57 ± 0.03 | 63.33 ± 0.02 | 61.97 ± 0.03 | 62.49 ± 0.02 |
| Average Performance | 61.38 ± 0.04 | 62.71 ± 0.04 | 61.87 ± 0.04 | 62.75 ± 0.04 | 63.70 ± 0.03 | 63.70 ± 0.03 |

Table 3: 27 Tasks Performance Comparison for the SIDER Dataset.

## A.2  SMILES-ONLY REPRESENTATION COMPARISON ON 6 DATASETS

| Classification | BBBP (↑) | | Bace (↑) | | Clintox (↑) | |
|---|---|---|---|---|---|---|
| | Smiles_only | ChemThinker | Smiles_only | ChemThinker | Smiles_only | ChemThinker |
| llama-3.1-8B | 67.73 ± 1.13 | 75.80 ± 0.95 | 75.16 ± 1.46 | 77.80 ± 2.97 | 99.68 ± 0.09 | 98.67 ± 2.11 |
| llama-3.1-8B-instruct | 69.22 ± 0.84 | 77.04 ± 1.00 | 75.32 ± 1.47 | 77.43 ± 1.89 | 99.33 ± 0.12 | 99.11 ± 0.42 |
| galactica-6.7b | 69.17 ± 1.12 | 74.94 ± 0.74 | 74.56 ± 1.86 | 78.88 ± 2.16 | 100.00 ± 0.00 | 99.93 ± 0.05 |
| galactica-30b | 68.57 ± 1.63 | 72.57 ± 1.32 | 77.37 ± 1.18 | 79.98 ± 2.67 | 99.99 ± 0.02 | 100.00 ± 0.00 |
| text-embedding-3-small | 71.54 ± 3.74 | 71.89 ± 1.36 | 75.96 ± 0.92 | 77.24 ± 1.65 | 99.64 ± 0.06 | 99.49 ± 0.07 |
| text-embedding-3-large | 71.65 ± 3.08 | 75.50 ± 1.26 | 77.31 ± 1.77 | 78.18 ± 0.96 | 99.86 ± 0.00 | 99.38 ± 0.09 |

Table 4: Classification results for BBBP, Bace, and Clintox datasets.

| Regression | ESOL (↓) | | FreeSolv (↓) | | Lipophilicity (↓) | |
|---|---|---|---|---|---|---|
| | Smiles_only | ChemThinker | Smiles_only | ChemThinker | Smiles_only | ChemThinker |
| llama-3.1-8B | 1.53 ± 0.03 | 0.49 ± 0.05 | 4.01 ± 0.73 | 2.27 ± 0.84 | 0.93 ± 0.02 | 0.74 ± 0.05 |
| llama-3.1-8B-instruct | 1.55 ± 0.03 | 0.44 ± 0.01 | 4.29 ± 0.93 | 2.01 ± 0.37 | 0.93 ± 0.02 | 0.73 ± 0.03 |
| galactica-6.7b | 1.69 ± 0.07 | 0.53 ± 0.25 | 4.58 ± 1.62 | 2.39 ± 1.39 | 0.89 ± 0.02 | 0.66 ± 0.02 |
| galactica-30b | 1.39 ± 0.02 | 0.55 ± 0.24 | 7.79 ± 1.34 | 2.22 ± 0.74 | 0.87 ± 0.03 | 0.73 ± 0.03 |
| text-embedding-3-small | 3.29 ± 0.09 | 0.50 ± 0.08 | 3.65 ± 0.23 | 2.43 ± 0.60 | 0.91 ± 0.01 | 0.77 ± 0.01 |
| text-embedding-3-large | 2.11 ± 0.10 | 0.48 ± 0.02 | 3.73 ± 0.83 | 2.35 ± 0.63 | 0.87 ± 0.01 | 0.79 ± 0.02 |

Table 5: Regression results for ESOL, FreeSolv, and Lipophilicity datasets.

## A.3 Example of Molecular Insights Report

We take the SMILES "CC(N)COc1c(C)cccc1C" as an example to generate a molecular insight report, showcasing the detailed analysis produced by our framework.

---

***General Question 1:***

How does the molecule's 3D shape of CC(N)COc1c(C)cccc1C change in different environments, and what are the effects of these changes?

***General Insights 1:***

CC(N)COc1c(C)cccc1C, is a substituted benzene derivative with the following structure:

**Benzene ring (c1c(C)cccc1C):** This part represents a benzene ring with two methyl groups (-CH$_3$) attached at the 1 and 4 positions, forming a 1,4-dimethylbenzene (p-xylene). The proximity of the two methyl groups (-CH$_3$) on the benzene ring can lead to steric hindrance, especially when bulky substituent (like the ethanolamine chain) are present. This steric hindrance might force certain bond angles or restrict free rotation, leading to specific 3D shapes that minimize energy.

**CC(N)CO:** This part represents a side chain, specifically an ethanolamine-like fragment (-CH(NH$_2$)-CH$_2$-OH) attached to the benzene ring via an ether bond (-O-). **Free Rotation around Single Bonds**: The -CH(NH$_2$)-CH$_2$-O- chain attached to the benzene ring can rotate freely around single bonds. These rotations lead to different **conformations** of the molecule, affecting how the side chain is positioned relative to the plane of the benzene ring.

---

***General Question 2:***

What are the key intermolecular forces that govern the behavior of this molecule, CC(N)COc1c(C)cccc1C, in various contexts?

***General Insights 2:***

The molecule contains both an amino group (-NH$_2$) and a hydroxyl group (-OH) in the ethanolamine side chain, which are capable of engaging in hydrogen bonding.

**In polar solvents** (e.g., water), the -NH$_2$ and -OH groups will form hydrogen bonds with water molecules, increasing solubility and influencing molecular conformation.

**In biological systems**, hydrogen bonding between these groups and amino acid residues in proteins or enzymes could be critical for molecular recognition, binding, and reactivity.

**In solid-state environments**, hydrogen bonding between adjacent molecules can stabilize the crystal structure, influencing physical properties such as melting point and solubility.

---

***General Question 3:***

How does the molecule, CC(N)COc1c(C)cccc1C, contribute to the overall chemical equilibrium in different contexts?

***General Insights 3:***

The molecule **CC(N)COc1c(C)cccc1C** contributes to chemical equilibrium as follows:

1. **In polar solvents**: Enhances equilibrium shifts toward dissolved species via hydrogen bonding with water.
2. **In non-polar solvents**: Reduces solubility, contributing minimally to equilibrium.
3. **In biological systems**: Participates in equilibrium through binding interactions with proteins/enzymes, influencing biological activity.
4. **In solid state**: Affects equilibrium by stabilizing crystal structures, impacting solubility and melting points.

Figure 5: Molecular Insights from General Molecular Thinking.

*Rule Generation from Literature Perspective:*

**Intuition-Driven Question:** Assume you are an experiment chemist, please come up with 20 rules that are important to predict if a molecule is blood brain barrier permeable (BBBP). The rules should be able to directly calculated from external tools or code libraries.

*Intuition  Insights:*

As an experimental chemist, here are 20 rules for predicting blood-brain barrier permeability (BBBP) that can be directly calculated using external tools or code libraries such as RDKit, ChemAxon, or ADMET software:
Physicochemical Properties

1. **LogP (Lipophilicity)**: Molecules with a LogP between 1 and 3 are more likely to permeate the BBB. This can be calculated using RDKit's MolLogP function.
2. **Molecular Weight**: Molecules with a molecular weight less than 500 Da have a higher chance of crossing the BBB. This can be calculated using RDKit's MolWt function.
3. **Topological Polar Surface Area (TPSA)**: Molecules with a TPSA less than 90 $\text{Å}^2$ are more likely to cross the BBB. This can be calculated using RDKit's TPSA function.
4. **Number of Hydrogen Bond Donors (HBD)**: Fewer than 3 hydrogen bond donors are preferred for BBB permeability. Use RDKit's NumHDonors function.
5. **Number of Hydrogen Bond Acceptors (HBA)**: Fewer than 6 hydrogen bond acceptors increase permeability. This can be calculated with RDKit's NumHAcceptors function.
6. **Number of Rotatable Bonds**: Fewer than 8 rotatable bonds improve permeability. RDKit's NumRotatableBonds can calculate this.
7. **Fraction of Sp3 Hybridized Carbons (Fsp3)**: A higher fraction of sp3 carbons (>0.25) increases BBB permeability. RDKit's CalcFractionCSP3 calculates this.
8. **Molar Refractivity**: Values between 40 and 130 are ideal for BBB permeability, which can be calculated using ChemAxon or RDKit's CrippenDescriptors.
9. **LogD at pH 7.4**: Molecules with a LogD between 1 and 3 at physiological pH tend to be more permeable. This can be predicted using ChemAxon or similar libraries.
10. **Solubility (LogS)**: Moderate aqueous solubility, typically LogS greater than -4, is favorable. This can be calculated using tools like ChemAxon's solubility predictor.Structural Properties
11. **Number of Aromatic Rings**: Molecules with 0–2 aromatic rings tend to cross the BBB more easily. RDKit's CalcNumAromaticRings can calculate this.
12. **Number of Nitrogen Atoms**: Fewer than 4 nitrogen atoms is ideal for BBB penetration. RDKit's NumAtomsMatchingQuery with nitrogen as the query can determine this.
13. **Number of Oxygen Atoms**: Fewer than 3 oxygen atoms are associated with better BBB permeability. RDKit can count oxygen atoms using NumAtomsMatchingQuery.
14. **Charge at pH 7.4**: Neutral or zwitterionic molecules are preferred for BBB permeability. Charge can be calculated using ChemAxon's pKa predictor or similar tools.
15. **Shape and Rigidity**: A low number of rotatable bonds (<8) and a relatively compact shape are favorable. Use RDKit's MolShapeDescriptors for estimation.
16. **Presence of Functional Groups**: The absence of highly polar functional groups such as carboxylates (-COOH) or sulfates ($-SO_4^{2-}$) improves BBB permeability. RDKit can identify these groups.ADME/Tox Properties
17. **Blood-Brain Barrier Prediction Index (BBB Score)**: Tools like ChemAxon or ADMET Predictor provide BBB scores, with higher scores indicating better BBB permeability.
18. **Permeability Coefficient (Pe)**: Calculated using specialized ADMET software, this measures the ability of a molecule to permeate through biological membranes, with higher values indicating better BBB permeability.
19. **P-glycoprotein Substrate Status**: Molecules that are not P-glycoprotein substrates are more likely to cross the BBB. This status can be predicted using tools like ADMET Predictor or RDKit.
20. **Lipinski's Rule of 5 Compliance**: Molecules that comply with Lipinski's rules (MW < 500, LogP < 5, HBD < 5, HBA < 10) are more likely to cross the BBB. Lipinski's rules can be calculated using RDKit or ChemAxon.

Figure 6: Molecular Insights from Intuition-Driven Thinking.

864
865
866
867
868
869
870
871
872
873
874
875
876
877
878
879
880

*Rule Observation from SMILES Subset:*
**Intuition-Driven Question:** Assume you are a very experienced biologist/Chemist. In the following data, with label 1, it means a molecule is blood brain barrier permeable (BBBP). With label 0, it means it is not. Please infer step-by-step to come up with 3 rules that directly relate the properties/structures of a molecule. The rules should be able to directly calculated from an external tool or code libraries. Please write 3 rules without explanation.

[*Sampled data points from training dataset.*]

*Intuition Insights:*
Here are three rules based on the provided data, focusing on properties/structures that can be calculated using external tools or code libraries:
1. **Rule 1**: Molecules with a **molecular weight** less than 350 Da are more likely to be blood-brain barrier permeable (label 1).(Can be calculated using RDKit's MolWt function)
2. **Rule 2**: Molecules with **fewer than 3 hydrogen bond donors** (HBD) are more likely to be blood-brain barrier permeable (label 1).(Can be calculated using RDKit's NumHDonors function)
3. **Rule 3**: Molecules with a **LogP value** between 1 and 4 tend to be blood-brain barrier permeable (label 1).(Can be calculated using RDKit's MolLogP function)

Figure 7: Molecular Insights from Intuition-Driven Thinking.

881
882
883
884
885
886
887
888
889
890
891
892
893
894
895
896
897
898
899
900
901
902
903
904
905
906
907
908
909
910
911
912
913
914
915
916
917

*Task Specific Question:*

Here is a SMILES formula: ([START_I_SMILES]) CC(N)COc1c(C)cccc1C ([END_I_SMILES])
Question: Will the chemical compound penetrate the blood-brain barrier?

[*Molecular Insights from General Molecular Thinking and Intuition-Driven Thinking*]

**Answer:**

*Task Specific Insights:*

**Answer**: Yes, the chemical compound is likely to penetrate the blood-brain barrier (BBB).
**Reasoning based on the Rules:**
1. **Molecular Weight**: The molecular weight of **CC(N)COc1c(C)cccc1C** is approximately 179.26 Da, which is well below the 350 Da threshold. This supports BBB permeability based on **Rule 1**.
2. **Hydrogen Bond Donors (HBD)**: The molecule contains one hydrogen bond donor (from the -$NH_2$ group), which is fewer than the limit of 3 HBDs. This aligns with **Rule 2**, indicating favorable permeability.
3. **LogP (Lipophilicity)**: The LogP value of the molecule is estimated to be between 2 and 3 (due to the presence of hydrophobic methyl groups and the aromatic ring), which falls within the optimal range for BBB penetration according to **Rule 3**.
**General Structure Analysis:**
- The presence of a small hydrophilic side chain (-$NH_2$ and -OH groups) balanced by a hydrophobic aromatic ring and methyl groups contributes to moderate lipophilicity, which enhances BBB permeability.
- The molecule lacks highly polar functional groups or an excessive number of rotatable bonds, both of which could impede passive diffusion across the BBB.

**Thus, based on the above rules and structural analysis, this compound is predicted to penetrate the blood-brain barrier.**

Figure 8: Molecular Insights from Task Specific Thinking.