# OpenReview forum: "ChemThinker: Thinking Like a Chemist with Multi-Agent LLMs for Deep Molecular Insights"
_ICLR.cc/2025/Conference — ICLR 2025 Conference Withdrawn Submission_

### Official Review · Reviewer_WwNT · 2024-10-30

**Soundness:** 1
**Presentation:** 2
**Contribution:** 2
**Rating:** 3
**Confidence:** 4

**Summary:**

I thank the authors for a really interesting read, extracting chemical insights from pre-trained LLMs is a great idea, and the amount of work done is clear to see.

Summary:
* The authors present the ChemThinker framework - a method to orchestrate large language models to extract meaningful insights
* They use three strands of investigation - one looking at general molecular properties, one on the task specific properties, and one looking at “intuition”
* The core idea is that LLMs pre-trained on huge quantities of high quality data often includes many scientific papers and textbooks, by prompting the models appropriately much of this information can be extracted for individual molecules / tasks.
* These textual properties are synthesised through a series of calls to the LLM to iteratively improve the outputs, and then these embeddings are combined and used as inputs to the final MLP layers for downstream tasks.
* Of particular interest was the component contribution analysis where the authors showed how different tasks relied on the different LLM pathways with different strengths,
* This paper shows an interesting approach to extracting the knowledge contained in LLMs for scientific work - as demonstrated here, this is clearly a promising avenue.

**Strengths:**

* Taking advantage of the knowledge contained in LLMs is a clear path for lots of scientific work and creating evaluations / proof of use cases is vital to increase adoption of this kind of approach.
* The analysis of the way in which different models / lines of investigation contribute to solving each task offers insight into how the tasks benefit relatively from factual information learned during the pre-training stage of the LLM vs numerical information obtained through RDKit.
* Evaluating over multiple backbones for the ChemThinker framework is good and starts to give insight into which models are strongest in different settings.

**Weaknesses:**

* The exact orchestration of LLM calls was hard to extract from the paper. I was unsure if the processes were a single string of pre-defined prompts to extract answers, or if as was suggested in Fig 1. There was more iteration involved to obtain the final answers.
* The questions shown (if they are the full extent of the prompts) could be improved / expanded. It prompts the question of why only 3? Why those three in that exact phrasing?
* The process was less transparent for the intuition driven thinking in 3.2 - I can tell the LLMs are given a persona to derive “rules” but no examples of this are shown. Given that these rules are used to control which RDKit features are generated this makes it hard to understand exactly what is being done here.
* I had a similar problem with the task specific thinking in 3.3 - It is not clear where the tailored insights to T are coming from / how that is being prompted of the LLM. An example is said to be in the appendix, but the appendix was missing from the PDF submitted to open Review.
* It is unclear the typical size of the total concatenated embedding vector that is extracted per molecule, the number of calls to each LLM to generate this, the prompt structure used for each “agent” and the size of the final MLP. Personally I would want to see these details to place the evaluation in context.
* The comparison to other work was fairly extensive - but due to the limited detail on the ChemThinker pipeline / model configuration this made the comparisons harder to understand.

**Questions:**

My main request is for more detail to be included in section 3 specifically to show the reader what the orchestration of the LLMs are actually doing. I would want to understand what kind of questions to ask to get the best performance (and how they were reached), how to get task specific insights and how to generate inputs for RDkit. Sections 3.1, 3.2, and 3.3 would be really useful, but in their current state are lacking enough information to adequately replicate any sort of results.

Further specific questions:

* Why were the results only shown for the top 2 backbones in the tables, I would have expected to see all results there to better understand the properties of each model
* Figure 3 was excellent - but I would like to see cleaner presentation, certainly flat bars if you think it should be a plot, but crucially the  lack of numerical values to extract made analysis of these results impossible. Maybe consider a table as I think this result is great!
* In the conclusion and introduction you make claims about providing insights into the predictions, it would be good to see more evidence of these insights. Perhaps these are included in the appendix and were missing in the submission.
* I also have questions about data leakage - many of the molecules in the dataset have well know properties and how much the LLMs are reporting the answers may not be reflective of performance on the benchmark. On one hand this doesn’t matter as the method is demonstrating how to extract this information from the models, but on the other the comparison against other SOTA models could be misleading in these cases. How do you think this could impact the results?
    * I had specific concerns in Table 1 for ClinTox where the results are reaching over 99.0, and in Table 2 for ESOL where the result of 0.44 is significantly lower than all other results. (NB: LLM4SD gets closest at 0.52)
    * More discussion on these results would add confidence to the results.

---

### Official Review · Reviewer_C1Nc · 2024-11-01

**Soundness:** 2
**Presentation:** 1
**Contribution:** 1
**Rating:** 1
**Confidence:** 4

**Summary:**

The authors propose a molecular property prediction method based on using and combinding LLMs' representations.

**Strengths:**

- The general topic of molecular property prediction and its interpretability is important.

**Weaknesses:**

# A) Lack of clarity and false and tenuous claims.

The work completely fails to embed itself into relevant related work. It provides a highly flawed and biased view on the field and misleads readers in many ways. The first paragraph cites completely inappropriate papers, such as (Yang, 2019) for "molecular property prediction". Molecular property prediction is a decade-old field an here, for example, papers by Corwin Hansch [1] should be cited. Neural network entered the field in the 1990s [2], and deep learning methods were used from 2014 on [3-6]. Toxicity prediction with deep learning methods started around 2015/2016 [6-7]. The authors should completely re-write the first paragraph and embed their work properly, and carefully assign credit to pioneering works in the field of molecular property prediction.

Als the second paragraph mentions "current approaches (Xia et al., 2022; Liu et al., 2022; Luo et al., 2024; Rollins et al., 2024)", which is unclear: does this mean "currently best performing methods" (in this case, the claim is false), or "current LLM-based approaches". Furthermore, current best-performing approaches for molecular property prediction are hybrid methods that combine deep neural networks on descriptors with GNN-based representations (like Yang, 2019) and those are not "statistical pattern" based. The authors should completely re-work the second paragraph to make its theme clear and cite appropriate works.

The authors claim that LLMs perform "reasoning" which is highly debated [8] and questionable. If the authors insist on the point that LLMs indeed perform reasoning, they should provided substantial evidence for that and also cite works critizing the reasoning capabilities of LLMs.

The authors claim that " [.fingerprint-based methods.] they normally rely on pre-defined fingerprint and may not fully capture the complex patterns [...]", while it is rather the other way around: ECFP-based methods are as expressive as 1-WL, while many GNNs are not. SMILES-based approaches often lose stereo-chemical patterns. The authors should rectify this false claim.


# B) Significance: The significance is very low due to inappropriate experiments and missing compared methods.

The authors perform molecular property only on eight tasks, while usually methods perform this on 1000s of tasks (e.g. [9,10]). Additionally, the MoleculeNet benchmarks are outdated and should not be used anymore [11]. There is also for sure an error in evaluation the ClinTox task because it is impossible to predict clinical toxicity of a molecule at 99.4 ROC AUC. Clinical toxicity is a very complicated end-point that depends on a lot of factors of the individual person, other medication, genetic composition, etc, such that even experimental replicates would never reach this quality. Reporting such high values at predicting clinical toxicity also leads to ethical concerns (see below).

The compared methods should also include several descriptor-based approaches. Since ChemThinker is an LLM-based approach, also other LLM-based method, such as KV-PLM [12] and CLAMP [13] should be compared.

Since the authors claim in the abstract that their method is interpretable, they should rather not compare predictive performance, but focus on interpretability metrics, e.g., whether the toxicity-inducing parts of the molecules could be identified.

# C) Originality:

Using LLMs and combining representations from different pre-trained models is already known. However, what could indeed provide some additional value is prompting LLMs to think about the general molecular structure.


# D) Technical errors:
The overall approach is ad-hoc. It is unclear why exactly these components should be put together in the suggested way. The authors should propertly motivate and justify their approach and design decisions. They should perform an ablation study to identify which components yield advances in predictive performance or interpretability.

It is unclear whether LLMs can properly handle the SMILES strings as input. Usually the tokenizer is inappropraite for molecules. A single molecule can have many different SMILES representations, which could lead to different internal representations of the LLMs. Furthermore, stereochemistry is often lost in the SMILES strings. The authors should perform SMILES augmentation to show that their method is invariant to permuting SMILES strings. The authors should also include several stereo-isomers to show that ChemThinker can meaningfully distinguis stereo-isomers.

The overall training objective is not mentioned. It is unclear whether only the MLP is trained or whether the whole architecture is fine-tuned. The authors should present their approach clearer, write down the objective function, considered and selected hyperparamters, training and implementation details.

Typos:
- Chapter 2 "Reltaed work"

References:
[1] Hansch, C., Maloney, P. P., Fujita, T., & Muir, R. M. (1962). Correlation of biological activity of phenoxyacetic acids with Hammett substituent constants and partition coefficients. Nature, 194(4824), 178-180.
[2] Huuskonen, J., Salo, M., & Taskinen, J. (1998). Aqueous solubility prediction of drugs based on molecular topology and neural network modeling. Journal of chemical information and computer sciences, 38(3), 450-456.
[3] Unterthiner, T., Mayr, A., Klambauer, G., Steijaert, M., Wegner, J. K., Ceulemans, H., & Hochreiter, S. (2014, December). Deep learning as an opportunity in virtual screening. In Proceedings of the deep learning workshop at NIPS (Vol. 27, pp. 1-9). Cambridge, MA.
[4] Dahl, G. E., Jaitly, N., & Salakhutdinov, R. (2014). Multi-task neural networks for QSAR predictions. arXiv preprint arXiv:1406.1231.
[5] Lusci, A., Pollastri, G., & Baldi, P. (2013). Deep architectures and deep learning in chemoinformatics: the prediction of aqueous solubility for drug-like molecules. Journal of chemical information and modeling, 53(7), 1563-1575.
[6] Unterthiner, T., Mayr, A., Klambauer, G., & Hochreiter, S. (2015). Toxicity prediction using deep learning. arXiv preprint arXiv:1503.01445.
[7] Mayr, A., Klambauer, G., Unterthiner, T., & Hochreiter, S. (2016). DeepTox: toxicity prediction using deep learning. Frontiers in Environmental Science, 3, 80.
[8] Wang, B., Yue, X., & Sun, H. (2023). Can ChatGPT defend its belief in truth? evaluating LLM reasoning via debate. arXiv preprint arXiv:2305.13160.
[9] Mayr, A., Klambauer, G., Unterthiner, T., Steijaert, M., Wegner, J. K., Ceulemans, H., ... & Hochreiter, S. (2018). Large-scale comparison of machine learning methods for drug target prediction on ChEMBL. Chemical science, 9(24), 5441-5451.
[10] Stanley, M., Bronskill, J. F., Maziarz, K., Misztela, H., Lanini, J., Segler, M., ... & Brockschmidt, M. (2021, August). Fs-mol: A few-shot learning dataset of molecules. In Thirty-fifth Conference on Neural Information Processing Systems Datasets and Benchmarks Track (Round 2).
[11] Walters, P. (2023). We need better benchmarks for machine learning in drug discovery. Practical Cheminformatics.
[12] Zeng, Z., Yao, Y., Liu, Z., & Sun, M. (2022). A deep-learning system bridging molecule structure and biomedical text with comprehension comparable to human professionals. Nature communications, 13(1), 862.
[13] Seidl, P., Vall, A., Hochreiter, S., & Klambauer, G. (2023, July). Enhancing activity prediction models in drug discovery with the ability to understand human language. In International Conference on Machine Learning (pp. 30458-30490). PMLR.

**Questions:**

See box above.

**Details Of Ethics Concerns:**

Reporting a ROC-AUC of 99.4% at predicting the clinical toxicity of a molecule can have severe consequences: such high values indicate that the predictions are reliable, and molecules predicted negative might be considered as safe, and non-toxic to humans and might be used in a clinical trial. Thus, potential harm might be done to humans if such incorrect high values performance values are reported.

---

### Official Review · Reviewer_vUho · 2024-11-02

**Soundness:** 2
**Presentation:** 2
**Contribution:** 2
**Rating:** 3
**Confidence:** 5

**Summary:**

The contribution introduces multi-agent LLM framework for chemistry that is intended to control the internal representations of concepts and functions within LLMs. The agents are initialized in such contexts that should emulate subject matter experts' approach to molecular analysis. The described agents offer insights into general molecular properties, data-driven analysis, and task-specific factors. Each perspective. Accumulated representations from the agents are processed by multi-layer perceptron (MLP).

**Strengths:**

I'd love to pinpoint a strength, I just can't find one.

**Weaknesses:**

The paper is poorly written. It is full of typos that should be caught by any spellchecker.

The paper claims to mirror the way of chemist's thinking. It fails to define the persona of a "chemist", though. This is important because chemistry is an exceptionally broad field that absolutely does not reduce to analysis of molecular structure.

What is "general molecular thinking" in Section 3.1? Traditional computational tools are absolutely going beyond calculating simple chemoinformatic descriptors - there's a huge field of quantum chemistry and molecular simulations that can handle really complicated chemical scenarios. The entire semantic structure of the section is so nebulous, it raises a question as of the authorship of the text. There's no single technical, reproducible, systematically improvable description of the respective agent.

The questions provided as examples barely makes sense for a trained SME. For example, "How does the molecule’s 3D shape change, and what are the effects of these changes?" cannot have a meaningful answer - molecules exist as Boltzman distributions of 3D configurations and the shifts of these distributions depend on the external conditions that have to be specified.

The same ambiguity and lack of technical substance characterizes sections 3.2 "intuition-driven thinking" and 3.3 "task-specific thinking".

Section 3.4 "thought-representation fusion" describes a  fusion procedure that exists completely out of context of the previous sections. Representations of molecular structure descriptors, such as SMILES, and representation of arbitrary reasoning utterances about SMILES are different entities and the authors clearly do not understand this.

The performance benchmarks suggest that the paper aims to solve simple supervised learning tasks in QSAR/QSPR domain. The performance sees some improvement compared to alternatives, but it should be noted that this moderate improvement is achieved by increasing computing footprint by the factor that equals the number of agents. This looks a lot like a case of diminishing returns.

Some of the data in the tables are incorrectly labeled to favor the reported model. For example, in Table 2, ESOL(1), performance of LLM4SD should be ranked better than ChemThinker (Gallactica).  There are several instances where comparison of the models in terms of average performance might look different from the comparison of the performance distributions (Table 1: BBBP ChemThinker OpenAI vs LLM4SD; SIDER ChemThinker OpenAI vs UniMol, etc)

One of the not-so-obvious aspects of this work is that while it is reporting regression and classification of molecules expressed in some electronic format, such as SMILES, it does employ LLMs that have full capability to help bad actors and bring harmful items into the physical world, by proposing synthetic routes. Unfortunately, the authors don't seem to realize that the simple feature-engineering task accomplished by LLM agents places their work in a much broader context that requires ethical guardrails.

**Questions:**

No questions, really. Please rewrite the manuscript with a) clear statement of the task you are solving, b) clear statement of the systematically improvable aspect of the modeling, and without a) nebulous verbiage about outstanding capabilities arising from LLMs in-context learning and multi-agency.

**Details Of Ethics Concerns:**

Here, LLM agents appear to be used to generate embedding of molecular descriptors. However, these are general-purpose LLMs that can be simultaneously used  by bad actors to propose ways of bringing harmful chemical entities into the world. The authors should address this possibility.

---

### Official Review · Reviewer_V6qb · 2024-11-04

**Soundness:** 2
**Presentation:** 2
**Contribution:** 2
**Rating:** 5
**Confidence:** 4

**Summary:**

ChemThinker leverages a multi-agent structure within large language models (LLMs) to control and interpret the internal representations of molecular concepts and functions. It mimics the way chemists analyze molecules by incorporating insights from three perspectives: general molecular properties, data-driven analysis, and task-specific factors. Each perspective functions as an agent, guiding the model’s internal representation to generate more interpretable and targeted predictions.

**Strengths:**

ChemThinker stands out for its application of a multi-agent framework within large language models (LLMs) to enhance interpretability in molecular property prediction. The approach draws inspiration from the analytical methods used by chemists, applying a novel multi-perspective representation structure to guide the model’s focus on relevant features. This method represents an expert knowledge within LLM capabilities, allowing for a highly targeted interpretability that mirrors real-world chemist reasoning. The integration of LLMs for molecular analysis through a structured, interpretative framework also pushes the boundaries of LLM application in cheminformatics.

**Weaknesses:**

The method proposed in ChemThinker does not introduce significant innovations compared to existing approaches. The use of LLM embeddings in conjunction with multi-layer perceptrons (MLPs) for molecular property prediction is not a novel concept.

The multi-agent framework is central to ChemThinker’s design, but the rationale behind choosing three specific perspectives—general molecular properties, data-driven analysis, and task-specific factors—could be clarified. Why were these perspectives selected, and are they mutually exclusive or interdependent in practice? The authors could include a discussion or ablation study examining the impact of each perspective on predictive performance and interpretability to validate their choices.

The experimental results presented in the paper do not demonstrate that ChemThinker achieves state-of-the-art performance in molecular property prediction across both classification and regression benchmarks. For a comprehensive comparison, the authors should consider the results reported in the following studies:

Ross, Jerret, et al. "Large-scale chemical language representations capture molecular structure and properties." Nature Machine Intelligence 4.12 (2022): 1256-1264.
Soares, Eduardo, et al. "A Large Encoder-Decoder Family of Foundation Models For Chemical Language." arXiv preprint arXiv:2407.20267 (2024).
These references provide significant insights into current methodologies and benchmark performances in the field. Additionally, incorporating the QM9 dataset into their experimental framework would enhance the robustness of their evaluation and allow for a more direct comparison with existing models that utilize this widely recognized benchmark in molecular property prediction.

**Questions:**

1. Can you provide a more detailed explanation of how the three perspectives (general molecular properties, data-driven analysis, and task-specific factors) are integrated within the multi-agent framework? Specifically, how do these perspectives interact to influence the LLM's internal representations?

2.  How does ChemThinker differentiate itself from existing models that also utilize LLM embeddings with MLPs? What specific contributions does your framework make that advance the field of molecular property prediction beyond previous approaches?

3. Why was the QM9 dataset not included in your experimental evaluations? Given its significance in the field, do you plan to evaluate ChemThinker on this dataset in future work?

4. In the results section, could you clarify how ChemThinker's performance compares to recent state-of-the-art models? To strengthen your claims of superior performance, consider including comparisons with a broader range of state-of-the-art models, particularly those referenced in the literature (e.g., studies by Ross et al. and Soares et al.).

---

### Note · Authors · 2024-11-20

I have read and agree with the venue's withdrawal policy on behalf of myself and my co-authors.